# Near field optical visualization of the nanoscale phase percolation dynamics of a VO$_2$ oscillator

Kajal Tiwari[1,2], Zhong Wang [1,2], Yishen Xie [1], Ajesh Kollakuzhiyil Gopi [1], Jae-Chun Jeon [1], Ke Xiao [1]✉ & Stuart S. P. Parkin [1]✉

Self-sustained resistance oscillation in vanadium dioxide (VO$_2$) are of significant interest for phase-based information encoding applications. However, the underlying mechanism behind the current-induced insulator-to-metal phase oscillation and its spatiotemporal dynamics remains elusive. Here, using high-resolution near-field optical imaging, we uncover distinct current-induced phase transition pathways in VO$_2$(001) thin films. We show that the formation of a persistent metallic patch within active region, defined as the area between the electrodes in a two-terminal model device serves as a prerequisite for oscillations. In this region, transient conductive filaments as narrow as 140 nm bridge the patch to the electrodes. Additionally, we observe oscillation modulated optical signals that extend well beyond the active region, providing clear evidence for a mechanism that would couple neighboring oscillators. Our work provides direct insight into the percolation dynamics that controls the oscillatory state of a VO$_2$ oscillator, paving the way to optimally designed oxide electronics.

Strongly correlated oxide materials, in which complex interactions between the lattice, electrons, and spin play a central role in the insulator-to-metal electronic phase transition, are of significant interest due to their potential for high-performance electronic devices[1,2]. In some of these materials, resistance levels can be easily controlled by several orders of magnitude by external perturbations such as electric field, temperature, strain, electrochemical potential, and optical excitation[3–7]. Among the wide range of correlated oxide materials, vanadium dioxide (VO$_2$) undergoes a reversible, first-order insulator-to-metal transition (IMT) with a substantial resistance change above room temperature, accompanied by a structural transformation from a monoclinic to a rutile tetragonal phase[8–10]. VO$_2$ is well known for exhibiting self-sustained electrical oscillations between insulating and metallic states when a constant current is applied within the negative differential resistance range[11]. This oscillatory behavior, observed across micro- and nanoscale VO$_2$ devices, is of particular interest for applications in spiking neuronal devices and coupled networks for

sensors and complex computational functionalities[12–16]. Recent experimental studies have demonstrated that coupling between nanoscopic VO$_2$ oscillators can be achieved via the exchange of thermal energy[17–19].

To date, the formation of static percolative pathways has been widely recognized as key to the abrupt resistance change in correlated oxide systems[20–23]. Efforts have been made to directly visualize the current-induced IMT within correlated systems using a variety of imaging techniques. Stationary filaments with widths of several micrometers have been spatially visualized by confocal microscopy on diamond-chip and nanobeams with NV-centers[24,25]. One particularly effective method is scattering-type scanning near-field optical microscopy (s-SNOM), which provides a non-contact and non-destructive imaging method to visualize the electronic and optical properties at the nanoscale[26–28]. s-SNOM has been successfully utilized to demonstrate temperature-dependent nanoscale heterogeneities associated with the IMT in VO$_2$ thin films and other correlated oxide systems[29–32].

[1]Max Planck Institute for Microstructure Physics, Halle, Germany. [2]These authors contributed equally: Kajal Tiwari, Zhong Wang.
✉e-mail: ke.xiao@mpi-halle.mpg.de; stuart.parkin@mpi-halle.mpg.de

Remarkably, to our best knowledge, no experimental studies have yet directly captured the spatiotemporal dynamics of filament formation within the oscillatory regime of $VO_2$ devices, leaving the underlying physical mechanism elusive.

Here, we uncover a variety of current-induced non-oscillatory and oscillatory responses in two-terminal $VO_2$ devices using cryogenic s-SNOM with a high spatiotemporal resolution (down to 3.3 ms per pixel and 20 nm). An oscillatory state is only observed after the emergence of persistent metallic patches (PeMPs) within the active region between the electrodes. These patches are formed only after a high enough current is initially applied. Subsequently, transient percolation paths are formed between the PeMPs and the electrodes, which induce resistance oscillations. Note that these PeMPs remain after the current is reduced. Furthermore, we observe oscillation-modulated s-SNOM sideband signals that extend up to 2 μm beyond the active region. These findings deepen our understanding of the microscopic mechanism of $VO_2$ resistance oscillations.

## Results

### Nano-imaging of current-driven phase transition

To investigate the current-induced IMT in $VO_2$, we fabricated our model devices from 10 nm thick $VO_2$ thin films grown on $TiO_2$ (001) substrates (refer to Supplementary Note 1 for structural characterization of the film). Rectangular Ru (2 nm)/Au (50 nm) electrodes were formed on the unpatterned films by photo-lithography and a lift-off technique. The electrodes were 4 μm wide and spaced 2 μm apart[6,8]. Variable temperature s-SNOM imaging was performed at an excitation frequency of 1000 cm$^{-1}$ in high vacuum ($10^{-6}$ mbar) (Fig. 1a). This excitation frequency has previously been demonstrated to show enhanced sensitivity to the intermediate state of the IMT in $VO_2$[29]. Further details can be found in the "Methods". Device characterization via resistance versus temperature measurements (Fig. 1b) shows a hysteretic change in resistance at 308 K during warming and 291 K during cooling, with a change that exceeds three orders of magnitude. For the s-SNOM experiments, the device was first cooled to 260 K and then warmed to the desired temperature of interest.

At $T_1 = 295$ K, within the hysteresis loop, the application of a threshold d.c current ($I_{(th)} = 48$ μA) led to a sharp drop in resistance. The threshold current was determined by gradually increasing the current in 1 μA steps until the transition occurred. Fourth order demodulated s-SNOM amplitude imaging, which captures the intrinsic optical contrast[27,28], revealed the formation of a large, isotropic, high reflectivity region spanning the region between the electrodes and beyond (Fig. 1c). It should be noted that this region did not revert to its original low reflectivity upon switching off the applied current until the device was cooled below the IMT temperature (291 K). The signal

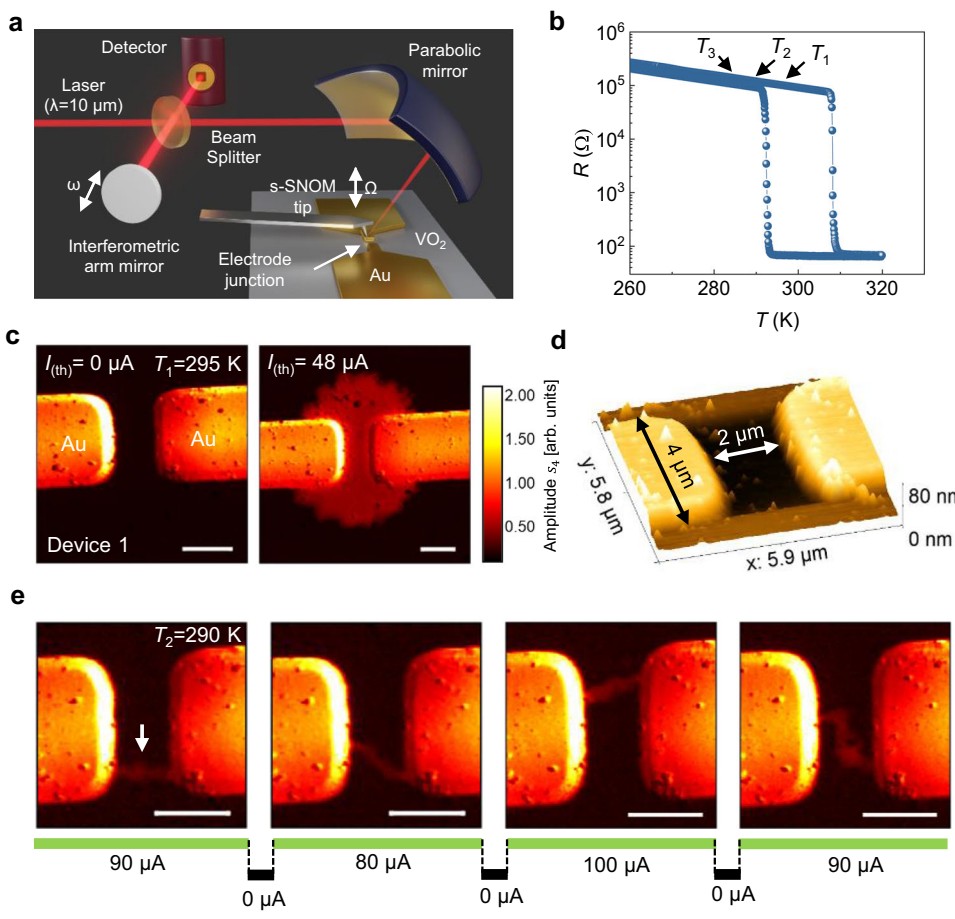

**Fig. 1 | s-SNOM imaging of current-induced phase transition in $VO_2$. a** Schematic of the experimental setup. A mid-infrared laser at a frequency of 1000 cm$^{-1}$ ($\lambda = 10$ μm) was used to probe the current-induced IMT and oscillations in the $VO_2$ device. The scattered light from the s-SNOM tip was collected via an MCT detector. The model devices under investigation have an active region of $4 \times 2$ μm$^2$. **b** $R$–$T$ curve showing a distinct hysteretic loop. The temperature of interest is marked as $T_1$, $T_2$, and $T_3$. **c** At $T_1 = 295$ K, 4th order demodulated s-SNOM amplitude images show the device before (left) and after (right) the application of the threshold constant current ($I_{(th)} = 48$ μA). A high reflectivity region is formed, which spans the entire active region and beyond. Scale bar: 2 μm. The color code of the s-SNOM amplitude is shown to the right, normalized to the Au electrode that is scanned simultaneously. **d** AFM height image of the device taken simultaneously with s-SNOM imaging. **e** At $T_2 = 290$ K, a volatile filament (marked by white arrow) assisted IMT is observed (leftmost image). Filament changes position on current off/on cycles, as shown in the images to the right. Scale bar: 2 μm. The same color code is used in (**e**) as in (**c**).

strength of this region matched with the intermediate metallic state of the temperature-driven IMT in the same film presented in the Supplementary Fig. 2, further confirming its metallic nature. All s-SNOM images were normalized to the simultaneously scanned Au electrodes to minimize alignment-related artifacts, and the same color scale was used unless stated otherwise. To verify that the observed change in optical reflectivity results from the IMT phase transition and not from topographic effects, which can significantly alter the optical contrast, we examined the topography simultaneously using variable temperature atomic force microscopy (AFM) that is integrated directly into the s-SNOM[33,34]. AFM imaging showed no topographic inhomogeneities in the high reflectivity region (see Fig. 1d). AFM phase and amplitude, which provide complementary information such as mechanical properties and chemical inhomogeneities, likewise show no discernible contrast (see Supplementary Fig. 11)[35,36].

At a lower temperature, $T_2 = 290$ K, a temperature that was chosen to be slightly below the IMT temperature, a higher threshold current $I_{(th)} \geq 80$ µA was required to induce a transition to the low resistance state. We observe a single narrow metallic filament bridging the two electrodes (Fig. 1e) instead of the aforementioned isotropic metallic region at the higher temperature $T_1$. These filaments disappeared when the current was removed and, for successive current applications, appeared at different positions and with different forms, as shown in several s-SNOM amplitude images in Fig. 1e. We find that the filaments did not follow the shortest path between the electrodes and appeared at different locations in each current cycle with no clear correlation to the current magnitude or the time interval between the current cycles. Notably, the laser power was maintained at a low enough level to prevent any thermal effect on the filament formation[24] (see Supplementary Note 13). The filament width, ranging between 300 to 500 nm, was substantially narrower than those reported in prior studies[24,25]. At intermediate temperatures between 290 K and 295 K, s-SNOM imaging (see Supplementary Fig. 3) reveals a mixture of filamentary features and metallic regions that depend on the current intensity and some of which remain when the current is switched off. This mixed-phase has, correspondingly, a variable resistance that we measured during the s-SNOM studies. This likely accounts for, for example, voltage-induced multiple resistance states that have been previously reported[37]. Despite the volatile nature of the filaments at 290 K, which, in principle, should provide favorable conditions for oscillatory behavior[38], we observe no oscillations at this temperature. In the following section, we investigate further the origin of the oscillatory state in VO$_2$.

## Spatial intricacies of the oscillatory state

Our electrical transport measurements reveal that the VO$_2$ devices exhibit oscillatory behavior only within the temperature range of 270 K to 285 K. A s-SNOM image of the insulating state of the device without applied current at $T_3 = 285$ K is shown in Fig. 2a. A step-wise increase of the applied current, in steps of 1 µA, eventually switches the device to a low resistance state (metallic state) at a threshold current $I_{(th)} \geq 130$ µA. This state comprises a PeMP between the electrodes with filaments extending from the PeMP to the electrodes creating a complete conducting channel as seen in Fig. 2b. This feature was consistently observed across all tested devices (see Supplementary Fig. 4). A current off/on (>130 µA) cycle results in the formation of a single filament on either side of the PeMP at a small number of sites (Fig. 2c). This contrasts with the much larger number of sites where we found the filaments formed at 290 K (Fig. 1e). Indeed in ~50 cycles each filament was mostly distinct. Furthermore, the filaments formed at 285 K are notably narrower than those formed at 290 K with widths ranging from 140 nm to 240 nm, as shown in the Supplementary Fig. 5. Following the formation of the PeMP above $I_{(th)}$, the device enters an oscillatory state when a current is applied within the range of 70 µA to 130 µA (see Fig. 2f). The inset to Fig. 2f shows a typical oscillatory $V(t)$ output of one

of these devices at a constant applied current. s-SNOM amplitude and phase images captured during these oscillations are shown in Fig. 2d. These images clearly show the formation of one or more dynamic filaments on either side of the PeMP (indicated by gray dashed boxes in the s-SNOM phase image) appearing at the same positions observed in the metallic phase at higher currents (Fig. 2b, c). Note that the s-SNOM signal was collected using a pseudo-heterodyne interferometric detection technique employing an integration time >3.3 ms, which is more than 10 times longer than the oscillation period of the device. The appearance of filaments at a fixed location during the oscillations suggests that these regions are preferentially involved in filament formation. These observations offer a natural explanation for why oscillations are absent near the IMT temperature: long, multi-site filaments lack the stability needed to support periodic dynamics. At lower temperatures, shorter and more strongly pinned filaments can nucleate and quench rapidly at fixed locations, enabling sustained oscillations. This framework also rationalizes the dependence on device size; smaller devices tend to oscillate at higher frequencies[17], possibly due to smaller filament dimensions and more localized nucleation. A practical implication is that actively stabilizing filament formation at a single site could positively influence the oscillation dynamics as observed in studies employing localized heat sources to anchor conduction channels[39].

When a current below 70 µA is applied, the device remains in a high-resistance state. Note that the PeMP persists even after the current is turned off while the filaments are absent, as shown in Fig. 2e. This explains the subtle drop in resistance observed after a high current application cycle, as shown in Supplementary Fig. 6. Notably, the PeMP shrinks upon cooling and eventually disappears below 100 K (see Supplementary Fig. 7).

To explore the filament dynamics, we performed a fixed-point temporal s-SNOM scan at a location where a filament formed (see Fig. 2g). A constant current was applied at the minimal threshold of the oscillatory regime, where oscillations are intrinsically unstable, to provide a suitable time window for resolving non-averaged signal changes. In this regime, we observed oscillations interrupted by intervals of inactivity, the duration of which can be both longer and shorter than our temporal resolution of 3.3 ms per pixel (see Supplementary Fig. 8a). As a result, the time trace of the s-SNOM signal fluctuates between two extrema. The minimum signal corresponds to the absence of a filament, representing either the insulating state during long inactivity or filament switching positions. The maximum signal reflects the transient filament during stable oscillations over the 3.3 ms window, giving the average signal of filamentary and non-filamentary states (see Supplementary Fig. 5c). Intermediate amplitudes arise from a mixture of oscillations and inactivity within one pixel, and the duration of inactivity within one pixel determines how far the signal shifts toward the minimum or maximum. Possible scenarios of filament formation that could lead to the observed time trace in Fig. 2g are shown in Supplementary Fig. 12. This intensity pattern supports the presence of ongoing filament reconfiguration with time. While our measurement protocol does not explicitly resolve how filament fluctuations correlate with oscillation timescales, it unequivocally demonstrates that filaments critically determine the device's low and high resistance states and exhibit self-sustained reconfiguration during resistance oscillation under constant current. Furthermore, the s-SNOM images reveal that the PeMP itself is pulsating with time. In a series of magnified images focusing on the PeMP edge shown in Supplementary Fig. 8, we observed intermittent disappearance and reappearance of boundary segments just below the threshold current of the stable oscillatory state. This indicates that the PeMP boundary is a dynamic entity. Such pulsating behavior aligns with expectations, given that the PeMP typically expands (shrinks) on heating (cooling) (see Supplementary Note 7). Consequently, thermal fluctuations during each oscillation cycle likely drive its expansion and contraction.

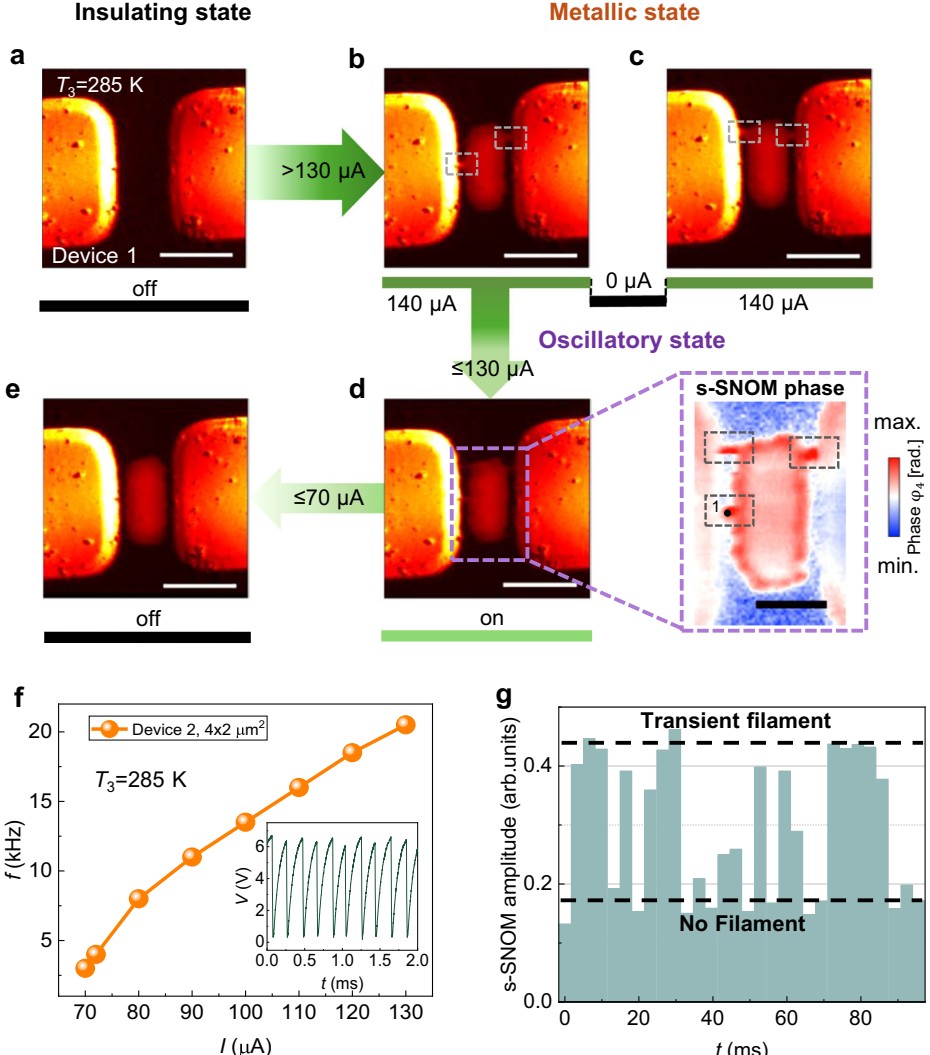

**Fig. 2 | Direct visualization of nano-percolation paths. a** s-SNOM amplitude image at $T_3 = 285$ K and $I = 0$ µA (OFF state, indicated below the image). The device is in an insulating state. Green arrows indicate applied current. Scale bar: 2 µm. **b** Above 130 µA, device turns metallic, accompanied by an emergence of a PeMP. Single filamentary extensions are seen from the PeMP connecting to the electrodes. **c** Filament reconfiguration occurs after a current off/on recipe. **d** Below 130 µA, the device now enters an oscillatory state. Transient filament positions as seen in the s-SNOM amplitude and phase images coincide with those found in the metallic state (**b**, **c**). Scale bar of the phase image: 1 µm. **e** Persistent nature of the metallic patch is evident from the image at zero applied current (insulating state), unlike the filaments, which are volatile in nature. **f** Oscillation frequency versus applied current. The inset shows a typical oscillation. **g** Bar chart of s-SNOM signal time evolution at the filament location (highlighted by a black dot at the location of filament 1 in the s-SNOM phase image of (**d**)), illustrating the stochastic flickering of filaments during unstable oscillations at the minimum threshold current for oscillation. All the data are taken from device 1 except for (**f**), which corresponds to device 2. Devices 1 and 2 are nominally identical.

This dynamic behavior, combined with the transient formation of filaments, constitutes the primary mechanism underlying the oscillatory state in our VO$_2$ devices.

To understand the non-volatility of the PeMPs and their suppressed IMT, we perform elemental mapping using energy-dispersive X-ray spectroscopy (EDX) in a $10 \times 5$ µm² device that had displayed a well-formed PeMP. Our EDX element mapping at ambient temperature reveals that the oxygen K-line intensity is reduced by approximately 4% within the PeMP (Fig. 3a, b). The corresponding s-SNOM and AFM height images are shown for comparison in the bottom panel of Fig. 3a. Line-cuts of the s-SNOM amplitude, AFM height and EDX are presented in Fig. 3b. The AFM height profile shows no variation across the PeMP, within experimental error, confirming the absence of significant topographical changes, while the s-SNOM profile at 285 K shows enhanced reflectivity in the same region, indicative of increased metallicity. AFM phase imaging reveals a subtle contrast at the PeMP site, consistent with our EDX results (see Supplementary Fig. 11). In our films, the oxygen vacancy channels are oriented along the crystal $c$-axis. Oxygen vacancies can be either introduced via ionic-liquid gating (electric-field) or by annealing in low oxygen pressures. These can induce a reduction in the transition temperature or, in some cases, a complete suppression of the transition, depending on the extent of oxygen loss[6,40,41]. To address this question, we performed COMSOL simulations to estimate the temperature and current density distribution in our devices in the insulating state. The simulated temperature distribution (see Fig. 3c) shows that current-induced heating is maximal in the central region and decreases towards the electrodes due to facilitated heat dissipation from the gold electrodes, as evident from the line-cut along the $x$-axis (line 2, purple) shown in Fig. 3d. This explains the spatial confinement that we found for the PeMP and the presence of an insulating region between the PeMP and the electrodes (see Fig. 2e). Details of the simulation and the current density distribution are discussed in Supplementary Note 9. Thus, we conclude that the PeMP is associated with a slight oxygen deficiency that is

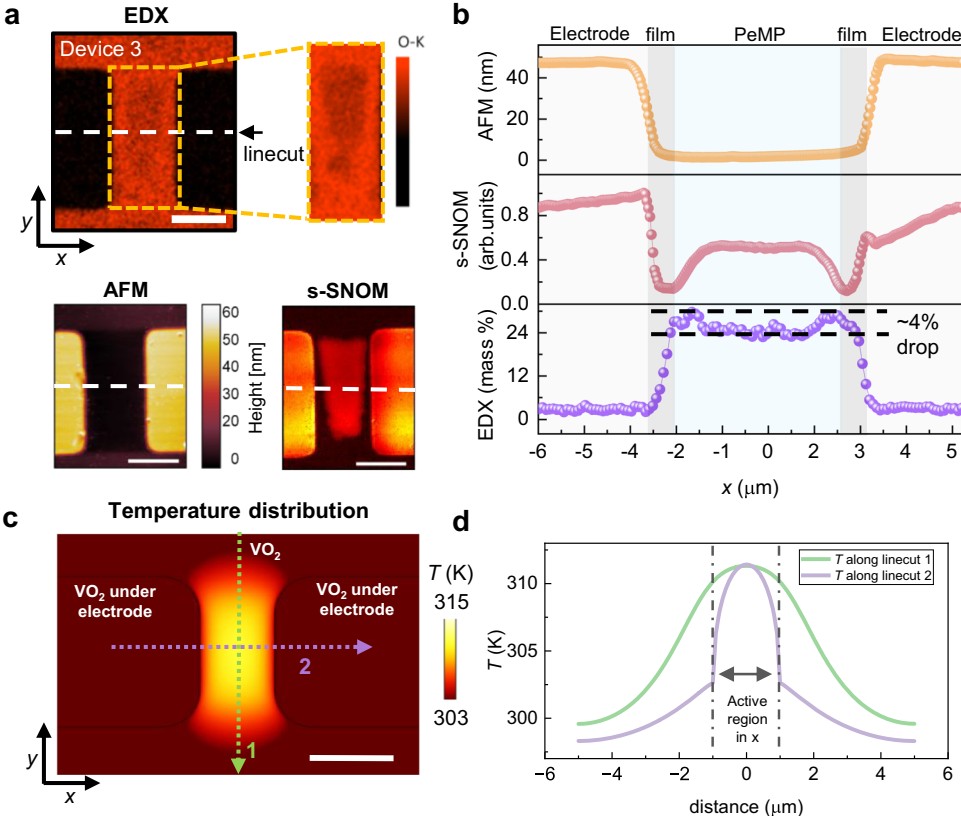

**Fig. 3 | Surface characterization of PeMP formation. a** EDX elemental mapping image for a larger device 3 with an electrode width of 10 μm and electrode spacing of 5 μm showing a decreased oxygen signal at the PeMP location. AFM and s-SNOM amplitude images of device 3 are presented for comparison. Scale bar: 5 μm. **b** Comparison of AFM height, s-SNOM, and EDX along the line-cut (white dashed line) of device 3. EDX shows a 4% decrease in oxygen content in the PeMP region (shaded in yellow color) while there is no loss of oxygen in the film area (dark gray color) on either side of the PeMP. **c** Simulated temperature distribution within a 4 × 2 μm² device. Scale bar: 2 μm. **d** Temperature distribution along the line-cut shown by line 1 (green) and 2 (purple) in (**c**). A sharp drop in temperature near the electrodes matches well with the presence of insulating regions between the PeMP and electrodes on either side.

formed from a localized, current-induced thermal process. We further employed Kelvin Probe Force Microscopy (KPFM) to investigate the electronic landscape of the PeMP, a technique capable of mapping the surface potential with sub-100 nm spatial resolution. KPFM has been demonstrated to effectively map the change of the work function during the temperature-dependent IMT[40]. Our KPFM results reveal a significantly elevated surface potential across the PeMP (Supplementary Fig. 10), suggesting a substantial difference in work function between the PeMP and the insulating region.

Based on our understanding of the PeMP's origin and dynamics, its behavior during the current induced IMT reveals a compelling analogy to the phenomenon of long-term potentiation (LTP) in biological systems, a fundamental mechanism in memory formation. Similar to LTP, where a single strong stimulation can cause a lasting change in synaptic strength, the PeMP retains its metallic state after initial stimulation and actively triggers oscillations and guides filament formation. In this analogy, electrodes serve as artificial neurons, while filaments function as synaptic links influenced by the PeMP. This coupled dynamic behavior suggests that PeMPs can operate as stable, memory-like nodes within neuromorphic systems.

## Oscillation modulated optical signal

So far, our investigation of the oscillatory state relies on spatial images, averaged over timescales 10 times longer than the oscillation period, giving an average temporal evolution. To directly probe the temporal characteristics of the VO$_2$ oscillations, we measure the full frequency spectrum of the reflected optical signal from s-SNOM. Here, the measurement was performed in homodyne mode by

turning off the oscillation ($\omega$) of the interferometric arm mirror (see Fig. 1a), so that the detector AC signal originates only from VO$_2$ resistance and s-SNOM tip oscillations, avoiding convolution in the frequency spectrum. When a current of 90 μA is applied with the tip in approach mode, we observed higher harmonics of optical signal spaced at intervals matching the device's oscillation frequency ($f_{VO_2} = 10.9$ kHz), which are significantly enhanced (purple curve in Fig. 4a) compared to those with tip retracted (blue curve in Fig. 4a), likely due to strong field confinement and enhancement at the tip apex[27,42]. Moreover, we observe VO$_2$ oscillation ($f_{VO_2}$) modulated multiple sidebands of the 1st harmonic s-SNOM signal at the tip-dither frequency ($f_{tip}$) (shown in Fig. 4a). The amplitude modulated sidebands are spaced at $f_s = nf_{tip} \pm mf_{VO_2}$ ($n, m = 1, 2, 3...$), clearly visible for $n = 1$ and up to $m = 6$. These sidebands disappear either when the tip is retracted or when the sample is not oscillating, indicating that they arise specifically from the oscillatory dynamics and have a spatial confinement. Among them, the 1st lower sideband was used to extract a local oscillation-modulated signal along the line-cut of the central region, as shown in Fig. 4b. The linecut position $y = 0$ μm corresponds to the center of the PeMP. Data have been normalized to lie in the range of [0, 1] for comparison of trends. The normalized sideband signal is maximized at the PeMP center and decreases as the tip is moved away from the center of the PeMP. The detectable sideband signals, which follow a Gaussian distribution (dashed gray curve), extend up to approximately 2 μm beyond the active region, revealing that the oscillatory dynamics affect even the insulating areas of the film outside the PeMP and filaments. The orange curve in Fig. 4b is the simulated temperature distribution

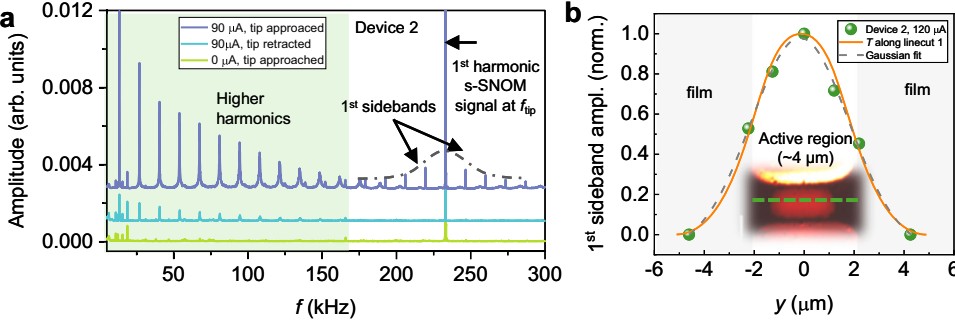

**Fig. 4 | Sideband characterization of VO₂ oscillation. a** Frequency spectrum of the reflected optical signal obtained on device 2 for three different scenarios: (1) Zero applied current and s-SNOM tip approached (green curve), (2) Ninety microampere and tip retracted (blue curve), (3) Ninety microampere and tip approached (purple curve), respectively. Sideband signals are indicated by the dashed envelope. 1st sideband signals are marked by arrows. **b** Normalized 1st sideband amplitude measured along the line-cut denoted by green dashed line in the inset image for an applied current of 120 µA. The line-cut position $y = 0\,\mu m$ corresponds to the center of the PeMP. The corresponding Gaussian fitting is shown as a gray dashed curve. Simulated temperature distribution (norm.) along the same line-cut is represented by the orange curve. Data have been normalized to lie in the range [0, 1].

(normalized) previously shown in Fig. 3d and matches well with our experimental observations.

## Discussion

In this work, we have uncovered the spatio-temporal nanoscale dynamics of the oscillatory state in a model VO₂ device with s-SNOM and found that a pulsating PeMP is required for oscillations; this thermally induced, oxygen-deficient region seeds flickering short filaments down to ~140 nm, defining a dynamic PeMP-filament landscape. Filaments are strongly pinned in the oscillatory regime, whereas higher temperatures allow for multi-site filaments, indicating that filament pinning is an essential factor for sustaining oscillations. In micron-scale junctions, the PeMP acts as a mesoscale bridge that lowers the filament nucleation barrier, while in more realistic nanoscale devices, oscillations may instead proceed through intrinsic filament formation without the need for a macroscopic thermally induced PeMP. Because the thermal landscape and the IMT pathway are strongly influenced by the substrate material[8,43], device design and size[17], a systematic study across different junction sizes and underlayers would be an important next step. Coupling transport with real-space infrared nano-imaging will help to establish how oscillation pathways scale, as well as yield design rules that reduce device-to-device variability and support robust, reproducible VO₂-based oscillator circuits for future oxide electronics.

## Methods

### Film growth and characterization

The VO₂ thin films were grown on TiO₂(001) substrates (CrysTec) by pulsed laser deposition (PLD) at 380 °C[6]. KrF laser pulses with ≈0.8 Jcm⁻² energy density were used to ablast a polycrystal VO₂ target (99.9%, Plasmaterials). The substrate-target distance was 5 cm. The growth atmosphere was 0.019 mbar O₂, and the sample was cooled down in 0.045 mbar O₂. XRD theta-2 theta scans for VO₂ thin films were performed with a Bruker D8 system with Cu-Kα line (1.54 Å wavelength).

### Surface, structural, and compositional characterization

AFM mappings in air were performed with a Bruker Dimension Icon system, ScanAsyst mode. KPFM was performed with an Asylum Research Cypher VRS system, SKPM/KPFM mode, with a doped Silicon tip (Bruker RTESPA-150). Scanning electron microscopy (SEM) imaging and EDX experiments were performed with a JEOL JSM-IT800 system. The EDX mapping was optimal when the beam voltage was 10 kV and the electron counting was 20 to 40 × 10³ counts per second.

### Electrical transport measurements

Resistance–temperature ($R$–$T$) relations were measured with the 2-point method on an Advanced Research Systems cryostat with a Keithley 2450 source meter. The temperature was ramped at 5 K min⁻¹, controlled by a LakeShore 336 temperature controller. Current–voltage ($I$–$V$) curves were measured on a cryogenic probe station. The current was supplied by a Keithley 6221 current source, and the voltage was measured by a Keithley 2182A nanovoltmeter. For the electric oscillations captured on a cryogenic probe station or in the s-SNOM chamber, the current was supplied by a Keithley 6221 current source and the waveform was captured by a 2-channel Agilent Technologies DSO5052A oscilloscope. Sideband optical amplitude measurements were carried out using MFLI from Zurich Instruments.

### Near field imaging

Near field (NF) imaging was performed using a cryogenic s-SNOM from NeaSpec of Attocube systems. Mid-infrared QCL laser (1000 cm⁻¹) was kept at a power below 0.8 mW to minimize sample heating. A platinum-iridium coated silicon tip with a radius of ~20 nm and a cantilever resonance frequency near 233 kHz (from NeaSpec) was used for scanning at a tapping amplitude of approximately 80 nm fixed for all the measurements. The tip temperature was fixed at 295 K. Scans were performed in high vacuum conditions with sample space pressure at 10⁻⁶ mbar. The reflected far-field signal was collected by the HgCdTe (MCT) detector. NF signal decays exponentially as the tip moves away from the sample surface. Therefore, to obtain a pure near-field signal from the large spurious background scattering from the tip shank, cantilever, and sample surface under direct illumination by the laser, the collected signal is demodulated to higher orders of the tip oscillation frequency. All the images presented in the manuscript pertain to the 4th-order tip demodulated s-SNOM signal. Remnant multiplicative background scattering is further eliminated through pseudo-heterodyne interferometric detection (interferometric arm mirror modulation by $\omega = 303$ Hz), which also enables simultaneous detection of the signal amplitude and the phase information via sideband frequency lock-in detection. Oscillation-modulated sideband signal extraction was performed in homodyne mode by turning off the oscillation of the interferometric arm. All images were normalized with an Au/Ru electrode scanned simultaneously to eliminate alignment-related artifacts that can cause signal variations. Image processing was done using the open-source software Gwyddion.

## Data availability

Data sets generated during the current study are available from the corresponding author on request.

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

## Acknowledgements

S.S.P.P. acknowledges funding from the International Max Planck Research School for Science and Technology of Nano-Systems and the Deutsche Forschungsgemeinschaft (DFG, German Research Foundation)—project no. 471731263. The Max Planck Center for Quantum Materials is gratefully acknowledged. K.T. thanks Georg Woltersdorf for guidance and support. Z.W. and K.X. thank Kai-Uwe Assmann for technical support for SEM and EDX. Z.W. thanks Guanmin Li for fruitful discussions.

## Author contributions

S.S.P.P. directed the project. K.T., Z.W., J.-C.J., K.X. and S.S.P.P. conceived the project and designed the experiments. Z.W. grew the films and performed XRD, *R–T*, AFM and KPFM characterizations. Y.X. fabricated the devices. K.T. performed s-SNOM and MFLI measurements. Z.W., J.-C.J. and K.T. measured the electrical oscillations. Z.W. and K.X. performed SEM and EDX experiments. K.T. and K.X. analyzed the SNOM results. A.K.G. helped with the MFLI analysis. K.T., K.X., Z.W., J.-C.J. and

S.S.P.P. prepared the manuscript. All authors discussed the results and participated in editing the manuscript.

## Funding

## Competing interests
The authors declare no competing interests.
