## [Transparent Peer Review File · Nature Communications]

Near field optical visualization of the nanoscale phase percolation dynamics of a VO₂ oscillator

Corresponding Author: Professor Stuart Parkin

Version 0:

Reviewer comments:

Reviewer #1

(Remarks to the Author)

“Near field optical visualization of the nanoscale phase percolation dynamics of a VO₂ oscillator” by Tiwari et al.

In this work, Tiwari et al., using scattering-type scanning nearfield optical microscopy (s-SNOM) observed the formation and percolation of persistent metallic patches (PeMPs) in a VO₂ thin film. The authors concluded that the PeMPs, is a prerequisite for the formation of transient conductive filaments. The flickering conductivity filaments potentially provides a mesoscopic explanation for the resistivity oscillation in VO₂, a key physical property that paves pathway towards neuromorphic computing devices.

The manuscript is overall well-written. The s-SNOM-based experimental techniques applied in this work are innovative. With proper supporting measurements such as EDX and KPFM and careful data analysis, the author provided a reasonable explanation for the observation of PeMPs and flickering filaments. The topic of this work fits the scope of Nature Communications well and the results presented are novel and could potentially impact multiple fields.

There are several questions that I hope the authors can address.

1. The metal-to-insulator transition in VO₂ is accompanied with structural transition. Through AFM topography map, the authors demonstrated no noticeable changes in topography like the ones observed in Ca₂RuO₄ or in other VO₂ films. However, are there any other responses in the mechanical signals in AFM scans? AFM probe amplitude and phase can be more sensitive to some mechanical features.
2. The flickering behavior of filament is an interesting and significant observation. However, the result presented in Fig. 2g are a little confusing. (1) There are several detection events with intermediate near-field scattering amplitude that are hard to be classified into “Filament” or “No Filament”. (2) If I count these intermediate states towards “Filament” state, then the statistics skews significantly towards “Filament” state. What’s the physical expectation here? A more careful statistical analysis would be helpful.
3. Following the questions above, the “on-and-off” behavior of conductive filaments does not show clear temporal patterns. With this random switching behavior, one would naively think that when averaging over multiple filaments, the temporal fluctuation would cancel out and the macroscopic sample should not show oscillation in time. I understand that the microscopic mechanism that connects the flickering filaments and resistivity oscillation is slightly beyond the scope of this work. However, as the authors claimed correlation between these two features, it would be helpful to discuss some intuitive physical pictures of possible scenarios.

Some minor comments:

1. The color scale bar in Fig. 1c is not well labeled with units. I assume the authors meant “S₄ (arb.u.)”, but please check and confirm. Similarly, the authors should also label the color bar in Fig. S3a for O₂A, otherwise the color bar makes little sense.
2. The experiment scheme presented in Fig 4a is confusing. The current form of schematic drawing depicts nothing but a standard s-SNOM setup in homodyne mode (as there is no reference arm shown in the drawing). Thus, I strongly

recommend revising this figure to include more detailed description of the sideband analysis setup.

Reviewer #2

(Remarks to the Author)

The paper by Kajal Tiwari/Zhong Wang is an impressive piece of work (Side remark: I support the two main authorship suggestion in such a complex interdisciplinary multimethod paper). It has been known that oscillations in VO₂ are related to the insulator metal transition, but to my knowledge no visualization has been demonstrated. Using s-SNOM in a model device is an excellent approach to study the behavior with high spatial and time resolution (though not on the time scale of the oscillations).

To my opinion, the line of argumentation is flawless, the experimental methods sophisticated and well chosen. The paper should be published.

To my opinion, there is only one thing missing which is a critical assessment of the model device. The abstract mentions that the authors pave the way "to optimally designed oxide electronics". The device, however, is a macro device. Therefore, the observation of a PeMP (persistent metallic patch) as a necessary ingredient to stable oscillations might simply be due to the large junction size. I would expect that any stable oscillatory behavior can only involve conducting regions on the nanoscale (as nicely observed in the paper). In a more realistic device setup, the VO₂ layer might be itself in the range of a few nanometers. Obviously, for such a more realistic device the here applied methods would not be suited. Therefore, I still believe that the paper reveals a key mechanism and should be published. But the impact on device design should be discussed in the paper in more detail and it should be pointed out that the device is a model device grown for the purpose of the experiment, but might be deposited on different underlayers (not TiO₂) and in different geometry later. The implication of the findings for such more realistic devices should be discussed (i.e. would a PeMP form in a nanolayers or would an artificial PeMP be required?).

Reviewer #3

(Remarks to the Author)

The authors displayed the interesting SNOM results for the phase-change material VO₂. The SNOM results for VO₂ in the oscillation states show that conductive filaments with narrow width forms in the active area. The SNOM signal is less investigated in the phase-change process. The authors show that this technique is powerful in the revealing the underlying physics. Other techniques such EDS, AFM, KPFM, are employed to reveal the microscopic details of the oscillating state. These findings enrich our understanding to the dynamics of VO₂ and would be interesting to researches of oxide electronics and beyond.

However, several issues need to be addressed prior to final publication at Nature Communications.

1) The phase change of VO₂ can be triggered by various stimuli, such as temperature, electric field and light. The authors presented SNOM results for VO₂ with increasing temperature (Fig. S2) and electric voltage (Figs. 1,2,3 et al.). A careful comparative study on the formation and dynamics of the filament evolution in the two processes should be made. This can improve our understanding on the transition mechanisms under different stimuli.

2) VO₂ phase is sensitive to the temperature. The authors made COMSOL analysis on the temperature profile. However, only results in arbitrary unit is presented (Fig. 3). Could the authors give more details on the physical model and numerical results? It would be beneficial to compare the evolution of high temperature (above critical temperature) region under current injection and the SNOM results.

3) The VO₂ is sensitive to any thermal fluctuation at the critical temperature. The authors stated that they have kept the laser power as low as possible to eliminate any perturbation. In SI, the power is given by 0.8 mW. Could the authors give more details on experiment parameters such as the spot size? It is suggested that authors make rigorous analysis to show that the heat effect of this light power can be ignored?

Version 1:

Reviewer comments:

Reviewer #1

(Remarks to the Author)

The authors have addressed the vast majority of comments. The revised version is suitable for publication in Nature Comm

Reviewer #2

(Remarks to the Author)

The authors have carefully addressed my requests. I agree with publication.

Reviewer #3

(Remarks to the Author)

The revised manuscript as well as the response has made response to my previous concerns. The present version is ready for publication and I therefore recommend acceptance of the manuscript.

Response to reviewers

Please note that throughout our responses (in blue) we have used the following color coding:

Brown color: Unaltered text in the revised manuscript.

~~Brown color:~~ Deleted text in the revised manuscript.

Green color: New text in the revised manuscript.

Reviewer: 1

In this work, Tiwari et al., using scattering-type scanning nearfield optical microscopy (s-SNOM) observed the formation and percolation of persistent metallic patches (PeMPs) in a VO₂ thin film. The authors concluded that the PeMPs, is a prerequisite for the formation of transient conductive filaments. The flickering conductivity filaments potentially provides a mesoscopic explanation for the resistivity oscillation in VO₂, a key physical property that paves pathway towards neuromorphic computing devices.

The manuscript is overall well-written. The s-SNOM-based experimental techniques applied in this work are innovative. With proper supporting measurements such as EDX and KPFM and careful data analysis, the author provided a reasonable explanation for the observation of PeMPs and flickering filaments. The topic of this work fits the scope of Nature Communications well and the results presented are novel and could potentially impact multiple fields.

Response: We sincerely thank the reviewer for considering our manuscript suitable for publication in Nature Communications and providing constructive suggestions for further improvement of our work. We are also grateful for the recognition of the novelty and potential broad impact of our findings on the VO₂ oscillations.

Please find below our detailed and point-by-point response to reviewer 1 comments:

1. The metal-to-insulator transition in VO₂ is accompanied by structural transition. Through AFM topography map, the authors demonstrated no noticeable changes in topography like the ones observed in Ca₂RuO₄ or in other VO₂ films. However, are there any other responses in the mechanical signals in AFM scans? AFM probe amplitude and phase can be more sensitive to some mechanical features.

Response: We thank the reviewer for raising this insightful question. Indeed, we did not observe any changes in the topography before or after the current induced insulator to metal transition. In systems such as Ca_2RuO_4 , as pointed out by the reviewer, the transition from the S' to L phase produces clear height changes due to c-axis expansion in the bulk crystal [1]. It has been reported that a slight topography/s-SNOM signal striation is observed in VO_2 microcrystals within the insulating state [2]. It is well established that the IMT in VO_2 is accompanied by a structural change from monoclinic insulating to rutile metallic phase, which for a thermally driven transition leads to a c-axis lattice contraction by 0.3% [3]. Given that our films are only ~ 10 nm thick, such a change is below the detection limit of AFM topography.

Figure R11: s-SNOM amplitude (O4A), AFM phase (M1P), amplitude (M1A) and height at **a**, 295 K and no applied current, **b**, 295 K and $I=48 \mu\text{A}$, **c**, 290 K and $I=90 \mu\text{A}$, **d**, 285 K and $I=140 \mu\text{A}$. Scale bar: 2 μm

Motivated by the reviewer's comment, we carefully revisited the AFM mechanical signals, specifically the AFM amplitude (M1A) and phase (M1P), in addition to the topography (height). These AFM images are obtained simultaneously with the s-SNOM images under vacuum. For the current induced IMT in VO₂ at temperature T₁ (295 K) and T₂ (290 K), we did not observe any noticeable changes in either M1A or M1P. However, at T₃ (285 K), where the PeMP forms during the current-induced IMT, we do observe a clear rise in the M1P signal at the PeMP site (Fig. R11). This observation is consistent with our EDX and KPFM results shown in Fig. 3 of the manuscript. Given that a change in chemical composition is known to result in AFM phase contrast [4], this observation further corroborates our interpretation.

To address the reviewer's comment, we have revised the manuscript accordingly and included the relevant data and discussion on the AFM amplitude/phase and topographic response during the current-induced IMT in the manuscript and Supplementary Materials with relevant references. In the revised manuscript:

Page 4, paragraph 1: AFM phase and amplitude, which provide complementary information such as mechanical properties and chemical inhomogeneities, likewise show no discernible contrast (see Supplementary Section S11) [4].

Page 6, paragraph 2: AFM phase image reveals a subtle contrast at the PeMP site, consistent with our EDX results (see Fig. S11).

In the revised Supplementary Materials, a new section S11 has been added to discuss AFM results in detail.

In summary, we have reexamined the AFM mechanical signals, including both phase and amplitude. We compared our results with the observed AFM signal in VO₂ and other materials and commented on the differences observed. Accordingly, we have revised the manuscript and added a new supplementary section S11 to present our data analyzed on AFM signals.

2. The flickering behavior of filament is an interesting and significant observation. However, the result presented in Fig. 2g are a little confusing. (1) There are several detection events with intermediate near-field scattering amplitude that are hard to be classified into "Filament" or "No Filament". (2) If I count these intermediate states towards "Filament" state, then the statistics skews significantly towards "Filament" state. What's the physical expectation here? A more careful statistical analysis would be helpful.

Response: We thank the reviewer for highlighting this important point and for carefully examining Fig. 2g. We apologize for any confusion caused by our original description and revisit our inference of the results in the main text.

We begin our answer by revisiting the experimental protocol used to study filament flickering. As stated in the manuscript, the time resolution of our s-SNOM point-scan is 3.3 ms. Since the intrinsic oscillation period is much shorter (see inset of Fig. 2f), each time window averages over ~10 oscillation cycles in the case of stable oscillations, resulting in a uniform signal in time without apparent flickering. To further investigate this, we performed additional comparative point-scan on a filament prone region of a $\text{VO}_2/\text{TiO}_2(001)$ device ($4 \times 2 \mu\text{m}^2$) at 285 K under stable oscillation conditions. As shown in Fig. R12a, the time trace under stable oscillation condition exhibits a constant s-SNOM amplitude with no apparent flickering. Specifically, at applied currents of $120 \mu\text{A}$ and $160 \mu\text{A}$, the SNOM signal have an average signal of 0.18 and 0.20 respectively, with fluctuations remaining below the noise level. In this regime, the s-SNOM signal of the oscillating state lies between those of the insulating and metallic states at the filament location, and the signal increases with oscillation frequency. However, due to averaging of the temporal fluctuation, if any, this method does not allow us to determine whether individual filaments appears or disappears spontaneously.

Figure R12a: Time trace of the s-SNOM signal in the point scan at the filament location at $50 \mu\text{A}$ (insulating state), $120 \mu\text{A}$ (stable oscillation), $160 \mu\text{A}$ (stable oscillation), $190 \mu\text{A}$ (metallic).

To overcome this averaging effect, we intentionally set the injected DC current at the minimum threshold of the oscillatory regime, where the oscillations are unstable. In this regime, we observe oscillations interrupted by random intervals of inactivity. Occasionally, the oscillations are interrupted by longer inactivity intervals (>3.3 ms) as shown in Fig. S8a. These irregular interruptions give rise to a non-uniform signal at the point of interest where the filament is likely to form (we have now highlighted the point of interest in Fig. 2d as shown in Fig. R12b and modified the figure caption accordingly).

Figure R12b: a, Revised Fig. 2d of the manuscript now highlights the point of interest probed in Fig. 2g by a black dot in dashed box 1. **b**, s-SNOM amplitude linecut at the transient filament location.

We have identified that our earlier inference in the Fig. 2g was misleading. In previous version of manuscript, we interpreted the maximum signal in Fig. 2g as corresponding to a “filament” and minimum signal as a “no filament” state. We have now corrected this here. The maximum signal should not be regarded as a ‘stable filament’ but a ‘transient filament’ resulting from stable oscillation within the 3.3 ms time window. In Fig. R12b, the line-cut across the transient filament in Fig. 2d of the manuscript (shown in the inset of Fig. R12b) reveals a maximum signal of 0.4 a.u., which is consistent with the maximum signal we obtained in Fig. 2g of the manuscript. The transient nature of the filament can also be inferred from the s-SNOM signal of a stable filament at the same location, measured when the device becomes metallic rather than oscillating under a current above the maximum oscillatory threshold. As shown in Fig. S5b, the same stable filament has a s-SNOM signal 0.65 a.u., while the surrounding region is around 0.15 a.u.. The mean of these two values closely matches the signal observed for the same filament in the transient state. Intermediate amplitudes in the time trace arise from a mixture of oscillations and inactivity within one pixel, where the duration of inactivity within one-pixel period determines how far the signal shifts toward the minimum or maximum. The

s-SNOM signal of the surrounding region is approximately 0.15 a.u., matching the minimum value in the time trace. This arises when no filament is present, which could be due to long inactivity exceeding 3.3 ms where the sample stays in the insulating state or simply that the filament switched positions. To provide some intuitive physical scenarios connecting filament flickering to macroscopic resistivity oscillations, one can consider the possibilities as shown in Fig. R12c.

Figure R12c: Possible scenarios of filament formation leading to the observed s-SNOM amplitude time trace for the point of interest (shown in yellow dashed box) and their possible relation to the corresponding voltage signals of the device. Transient filament is denoted by light shaded line while stable/non-transient filament is dark shaded line.

As shown in the figure, the s-SNOM signal labeled ‘1’ corresponds to the absence of a filament, as it matches the no-filament value (~0.15 a.u. in Fig. R12b). Possible scenarios at the probed location (yellow dashed box) include A-F, where stable filaments are indicated by solid orange lines and transient filaments by translucent orange lines. Among these, scenario B is likely the case. Signal ‘2’, which matches the transient-filament value (~0.4 a.u. in Fig. R12b), could arise from scenarios G or H, with scenario G, involving transient filaments on both sides, being more consistent. Signal ‘3’ lies at an intermediate level and is a mixture of scenarios of signal

1 and 2. Signal '4' slightly exceeds the transient-filament level and is consistent with scenarios I or J, where the probed filament exhibits a more stable character. However, since the fully stable filament signal reaches ~ 0.65 a.u., this minor increase of ~ 0.03 a.u. is more likely within experimental uncertainty rather than a change in filament behavior. The corresponding device voltage-time traces and the filament state they likely represent are illustrated on the right side of the Fig. R12c.

We have revised the manuscript to provide a better and correct explanation of the observed flickering in Fig. 2g and the physical origin of the intermediate values and corrected the Fig. 2g label from 'filament' to 'transient filament'. In the revised manuscript, page 6, paragraph 1: ~~To explore the filament dynamics, we performed a fixed-point temporal s-SNOM scan at a location where a filament formed (Fig. 2g). A constant current was applied at the minimal threshold of the oscillatory regime to sustain oscillations intercepted with prolonged non oscillatory (high resistance) intervals, thereby providing a suitable time window to resolve non averaged signal changes.~~ where oscillations are intrinsically unstable, to provide a suitable time window for resolving non-averaged signal changes. ~~With a temporal resolution of 3.3 ms per pixel, the time trace shows fluctuations between two extrema, corresponding to the filamentary and non filamentary state s-SNOM signals, respectively~~ In this regime, we observed oscillations interrupted by random intervals of inactivity, the duration of which can be both longer and shorter than our temporal resolution of 3.3 ms per pixel (see Fig. S8a). As a result, the time trace of s-SNOM signal fluctuates between two extrema. The minimum signal corresponds to the absence of a filament, representing either the insulating state during long inactivity or filament switching positions. The maximum signal reflects the transient filament during stable oscillations over the 3.3 ms window, giving the average signal of filamentary and non-filamentary states (see Fig. S12b). Intermediate amplitudes arise from mixture of oscillations and inactivity within one pixel, and the duration of inactivity within one pixel determines how far the signal shifts toward the minimum or maximum. Possible scenarios of filament formation that could lead to the observed time trace in Fig. 2g are discussed in Supplementary Section S12. This intensity pattern supports the presence of ongoing filament reconfiguration with time. While our measurement protocol does not explicitly resolve how filament fluctuations correlate with oscillation timescales, it unequivocally demonstrates that filaments critically determine the device's low- and high-resistance states and exhibit self-sustained reconfiguration during resistance oscillation under constant-current.

The revised caption of Fig. 2g now reads: **g, Bar chart of s-SNOM signal time evolution at the filament location** (highlighted by a black dot at the location of filament 1 in the s-SNOM phase image of Fig. 2d) illustrating the stochastic flickering of filaments during ~~the oscillations~~ unstable oscillations at the minimum threshold current for oscillation.

We have revised our Supplementary materials and added a new section S12 which summarizes the abovementioned discussion.

In summary, we have clarified the experimental protocol used to study filament reconfiguration, performed additional measurements in the stable oscillation state to emphasize the rationale behind our approach, corrected and elaborated on the origin of the intermediate values in Fig. 2g, and included a discussion of the possible scenarios, supported by illustrative schematics, to clarify the relationship between filament flickering and device resistivity oscillations. We have also revised the manuscript accordingly and added a new Section S12 in the Supplementary Materials to summarize these discussions.

3. Following the questions above, the “on-and-off” behavior of conductive filaments does not show clear temporal patterns. With this random switching behavior, one would naively think that when averaging over multiple filaments, the temporal fluctuation would cancel out and the macroscopic sample should not show oscillation in time. I understand that the microscopic mechanism that connects the flickering filaments and resistivity oscillation is slightly beyond the scope of this work. However, as the authors claimed correlation between these two features, it would be helpful to discuss some intuitive physical pictures of possible scenarios.

Response: We thank the reviewer for this insightful question and agree that temporal fluctuations would indeed cancel out, resulting in no change in the s-SNOM signal over time. As discussed in our response to Q2, we have revisited our experimental protocol and performed additional measurements in the stable oscillation regime to clarify our choice of measuring in the skipping oscillation regime. In this regime, the temporal fluctuations are aperiodic and therefore do not cancel out, offering an opportunity to study self-induced filament flickering. We have also included a clear schematic illustrating the possible physical scenarios in Fig. R12c. We hope this addresses the reviewer’s question to the best of our ability.

Some minor comments:

1. The color scale bar in Fig. 1c is not well labeled with units. I assume the authors meant “S4 (arb.u.)”, but please check and confirm. Similarly, the authors should also label the color bar in Fig. S3a for O2A, otherwise the color bar makes little sense.

Thank you for the correction. The values are indeed in arbitrary units, and this has now been fixed.

2. The experiment scheme presented in Fig 4a is confusing. The current form of schematic drawing depicts nothing but a standard s-SNOM setup in homodyne mode (as there is no reference arm shown in the drawing). Thus, I strongly recommend revising this figure to include more detailed description of the sideband analysis setup.

Response: We thank the reviewer for the comment and apologize for any confusion caused by the original schematic and lack of explanation.

The schematic in fig. 4a of the manuscript depicts our s-SNOM setup in homodyne mode, without the slow oscillation of the interferometric reference arm mirror. In our measurements presented in Fig.4, the sideband arises purely from the modulation of the optical signal due to VO₂ oscillations combined with the tip-tapping frequency of the SNOM. We intentionally did not use pseudo-heterodyne detection for sideband detection to avoid the complication of mixing two low-frequency signals, which produced a convoluted envelope around the tip-tapping frequency. While not using pseudo-heterodyne mixes amplitude and phase information, here we focus only on the intensity. We have verified that the analyzed signal has some level of spatial resolution (signal changes with the tip location as seen in Fig. 4c) and shows no background contribution (signal disappears when tip is retracted, see Fig. 4b) at the lower 1st sideband of the tip-tapping frequency combined with the VO₂ oscillation.

For clarity, we removed Fig. 4a, replaced Fig. 1a with a full schematic of the s-SNOM setup working in the pseudo-heterodyne mode and modified the text in the manuscript to indicate that the s-SNOM is being used in the homodyne mode for results presented in Fig. 4.

In the revised manuscript, page 8, paragraph 1: Here, the measurement was performed in homodyne mode by turning off the interferometric arm mirror oscillation (see Fig. 1a), so that the detector AC signal originates only from VO₂ resistance and s-SNOM tip oscillations, avoiding convolution in the frequency spectrum.

In summary, we have revised the experimental schematic and the manuscript to clearly distinguish the homodyne mode used here for VO₂ sideband analysis from the pseudo-heterodyne mode used in our previous measurements.

Reviewer: 2

The paper by Kajal Tiwari/Zhong Wang is an impressive piece of work (Side remark: I support the two main authorship suggestion in such a complex interdisciplinary multimethod paper). It has been known that oscillations in VO₂ are related to the insulator metal transition, but to my knowledge no visualization has been demonstrated. Using s-SNOM in a model device is an excellent approach to study the behavior with high spatial and time resolution (though not on the time scale of the oscillations).

To my opinion, the line of argumentation is flawless, the experimental methods sophisticated and well chosen. The paper should be published.

To my opinion, there is only one thing missing which is a critical assessment of the model device. The abstract mentions that the authors pave the way “to optimally designed oxide electronics”. The device, however, is a macro device. Therefore, the observation of a PeMP (persistent metallic patch) as a necessary ingredient to stable oscillations might simply be due to the large junction size. I would expect that any stable oscillatory behavior can only involve conducting regions on the nanoscale (as nicely observed in the paper). In a more realistic device setup, the VO₂ layer might be itself in the range of a few nanometers. Obviously, for such a more realistic device the applied methods would not be suited. Therefore, I still believe that the paper reveals a key mechanism and should be published. But the impact on device design should be discussed in the paper in more detail and it should be pointed out that the device is a model device grown for the purpose of the experiment but might be deposited on different underlayers (not TiO₂) and in different geometry later. The implication of the findings for such more realistic devices should be discussed (i.e would a PeMP form in a nanolayers or would an artificial PeMP be required?).

Response: We thank the reviewer for their thoughtful comments and constructive suggestions. We are especially grateful for the recognition of the interdisciplinary and multimethod nature of our work, and we appreciate your support for the shared first authorship. We are encouraged by your recommendation for publication.

To address the impact of device design and substrate choice, we have expanded the discussion to clarify how these factors can influence the IMT in general, supported by relevant literature: IMT in VO₂ is highly sensitive to the choice of substrate and its crystallographic orientation. For instance, VO₂/TiO₂ (001) exhibits a reduced transition temperature (~308 K) compared to VO₂/Al₂O₃ (~340 K) [5]. VO₂/TiO₂ (110) shows a higher transition temperature (~369 K) due to strain effects [6]. Moreover, geometry, layout and scaling critically affect the thermal landscape and oscillatory behavior, with smaller devices generally exhibiting higher oscillation frequencies and lower threshold currents [7]. Nanoscale manipulation of VO₂/TiO₂(001) thin-film device via carbon nanotube acting as a local heat source created anchored conduction channel leading to an enhancement of the spiking frequency [8, 9]. Devices with an electrode gap as small as 20 nm has been reported to exhibit a single resistance jump across the IMT, indicative of non-percolative, single-domain switching behavior [10]. These insights emphasize that substrate properties and device geometry collectively play crucial roles in determining the IMT behavior and oscillatory dynamics in VO₂ based devices. Future studies should systematically explore how these factors influence the IMT, extending from the present micrometer-scale structures to nanodevices. Investigations of VO₂ films with varying thicknesses and substrates of differing strain and thermal conductivity will be essential to establish design principles for nanoscale oxide electronics.

Regarding the reviewer's question on the fate of PeMPs in smaller devices, s-SNOM tip that we used cannot resolve below ~40 nm and this method is not suitable for vertical VO₂ nano-junctions [11], we fabricated planar devices with 200 nm electrode gap and 400 nm width. However, due to the ESD sensitivity of these junctions, electrical measurements in our commercial s-SNOM setup were not feasible and we could not yet resolve this issue. We strongly agree that exploring the conducting channels active in nanoscale geometries is an important future step.

Based on our observations and literature, we infer that while PeMPs play a key role in stabilizing oscillations in micron-scale devices, oscillations in nanoscale junctions may proceed through single filament formation without requiring a macroscopic thermally induced PeMP. Furthermore, following the recommendation of Reviewer 3 (comment 1), we carefully re-evaluated our results on temperature evolution of current induced IMT: In the temperature range that supports oscillations, filaments are strongly anchored, whereas at higher temperatures they extend into multi-site configurations that do not sustain oscillations. Our critical assessment of these observations leads us to conclude that localized filament pinning is

a necessary factor governing the system's ability to oscillate within a specific temperature window. The corresponding discussion has been incorporated into the revised manuscript (see page 5 and the conclusion section).

We have revised the conclusion of the manuscript to explicitly discuss the role of the PeMP and its implications for smaller or more realistic nanoscale devices. In the revised manuscript, page 8-9: In this work, we have uncovered the spatio-temporal nanoscale dynamics of the oscillatory state in a model VO₂ device with s-SNOM and found that a pulsating persistent metallic patch (PeMP) is required for oscillations; this thermally induced, oxygen-deficient region seeds flickering short filaments down to ~140 nm, defining a dynamic PeMP-filament landscape. Filaments are strongly pinned in the oscillatory regime, whereas higher temperatures allow multi-site filaments, indicating that filament pinning is an essential factor for sustaining oscillations. In micron-scale junctions, the PeMP acts as a mesoscale bridge that lowers the filament nucleation barrier, while in more realistic nanoscale devices, oscillations may instead proceed through single filament formation without the need for a macroscopic thermally induced PeMP. Because the thermal landscape and the IMT pathway are strongly influenced by the substrate material [5, 6], device design and size [7], a systematic study across different junction sizes and underlayers would be an important next step. Coupling transport with real-space infrared nano-imaging should establish how oscillation pathways scale and yield design rules that reduce device-to-device variability and support robust reproducible VO₂ based oscillator circuits for future oxide electronics.

We have also clarified throughout the text that our current devices are model systems grown for the purpose of this study. The manuscript has been revised accordingly, and we now clearly state that the device is a model device.

In the revised manuscript: page 3: To investigate the current-induced IMT in VO₂, we fabricated our model devices from 10 nm thick VO₂ thin films grown on TiO₂ (001) substrates (refer to supplementary section S1 for structural characterization of the film).

Page 8, conclusion section: In this work, we have uncovered the spatiotemporal nanoscale dynamics underlying current-induced metal–insulator transitions and oscillations in a model device of VO₂.

The revised caption of Fig. 1 of the manuscript: The model devices under investigation have an active region of ~4 x 2 μm².

In summary, we have incorporated a more detailed discussion of how device design and substrate influence IMT and oscillations, clarified the model nature of the studied device, and revised the conclusion to contextualize our findings and highlight directions for future nanoscale studies.

Reviewer: 3

The authors displayed the interesting SNOM results for the phase-change material VO₂. The SNOM results for VO₂ in the oscillation states show that conductive filaments with narrow width forms in the active area. The SNOM signal is less investigated in the phase-change process. The authors show that this technique is powerful in revealing the underlying physics. Other techniques such EDS, AFM, KPFM, are employed to reveal the microscopic details of the oscillating state. These findings enrich our understanding to the dynamics of VO₂ and would be interesting to researches of oxide electronics and beyond.

We thank the reviewer for their thoughtful comments and constructive suggestions. In response, we have carefully revised the manuscript to address the concerns raised. Please find below a point-by-point response to the reviewer 3 comments:

1. The phase change of VO₂ can be triggered by various stimuli, such as temperature, electric field and light. The authors presented SNOM results for VO₂ with increasing temperature (Fig. S2) and electric voltage (Figs. 1,2,3 et al.). A careful comparative study of the formation and dynamics of the filament evolution in the two processes should be made. This can improve our understanding on the transition mechanisms under different stimuli.

Response: We thank the reviewer for the thoughtful suggestions and for highlighting our key results. As noted, the IMT in VO₂ can be triggered by various stimulus such as temperature, electric field and light. In VO₂ thin films, increasing temperature induces a percolative network of metallic islands (Fig. S2), as previously demonstrated by Qazilbash et. al. using s-SNOM [12, 13]. Light can also trigger an IMT, as demonstrated by Sternbach et. al. where they revealed heterogeneous nucleation at picoseconds timescale using a pump-probe s-SNOM [14].

For current-induced IMT in VO₂, prior studies have established a filamentary transition pathway, for instance, Feng et. al. imaged laser-controlled formation of a single stationary filament of width 9.7 μm in an electrode gap of 50 μm and width 100 μm [15]. An optical

reflectivity study by Del Valle et. al. on filament nucleation and time evolution at temperatures well above the oscillatory threshold hinted that higher voltages may produce narrower, faster-nucleating filaments due to current focusing into smaller hotspots [16]. Furthermore, nanoscale manipulation of VO₂/TiO₂(001) thin-film device via carbon nanotube acting as a local heat source created anchored conduction channel leading to an enhancement of the spiking frequency [8, 9]. Building on this, we sought to understand two underexplored issues:

- How does the temperature at which the sample is held influence a current-induced IMT?
- What are the conducting elements during resistance oscillations observed far below the thermal IMT temperature.

These two questions are closely related. Because resistance oscillations emerge at lower temperatures, the relevant physics concerns not only the formation of conducting elements but also their dynamics and the underlying factors that govern them.

Figure. R31: Scenarios of current-induced IMT at different temperatures. Inset shows the Fig. 1b in the main text: R-T curve showing a distinct hysteretic loop.

We first summarized our experimental results of current-induced IMT at representative temperatures as shown in Figure R31, we find:

- Near the IMT temperature (within the hysteresis regime): the current-induced IMT proceeds via the emergence of a large metallic patch.
- Slightly below, yet within the hysteresis: the transition is mediated by a broader, branched filamentary structure (see Supplementary section S3).
- Just below the hysteresis window: a single filament forms, but its nucleation site is stochastic, and its path can be non-linear.
- In the oscillatory regime (well below the IMT temperature): we observe a PeMP + filament system. Relative to higher-temperature filaments, this filament is shorter, narrower, and reproducibly nucleates at two preferred sites (in our model device).

These observations offer a natural explanation for why oscillations are absent near the IMT temperature: long, multi-site filaments lack the stability needed to support periodic dynamics. At lower temperatures, shorter and more strongly pinned filaments can nucleate and quench rapidly at fixed locations, enabling sustained oscillations. This framework also rationalizes the dependence on device size; smaller devices tend to oscillate at higher frequencies possibly due to smaller filament dimensions and more localized nucleation. A practical implication is that actively stabilizing filament formation at a single site could positively influence the oscillation dynamics.

To better emphasize our interpretation of the results, we have added the abovementioned discussion to the revised manuscript on page 5 as follows: These observations offer a natural explanation for the absence of oscillations near the IMT temperature: long, multi-site filaments lack the stability required to support periodic dynamics. At lower temperatures, shorter and more strongly pinned filaments can nucleate and quench rapidly at fixed locations, enabling sustained oscillations. This framework also rationalizes the dependence on device size: smaller devices tend to oscillate at higher frequencies possibly due to smaller filament dimensions and more localized nucleation. A practical implication is that actively stabilizing filament formation at a single site could positively influence the oscillation dynamics as observed in studies employing localized heat sources to anchor conduction channels [9].

In summary, we have revisited the inference deduced from our observations and came up with a careful interpretation of our results on the temperature dependence of the current-induced

IMT in our model two-terminal VO₂ device. We have updated the manuscript by adding the fine details as discussed above.

2. VO₂ phase is sensitive to the temperature. The authors made COMSOL analysis on the temperature profile. However, only results in arbitrary unit is presented (Fig. 3). Could the authors give more details on the physical model and numerical results? It would be beneficial to compare the evolution of high temperature (above critical temperature) region under current injection and the SNOM results.

Response: We thank the reviewer for the comment and apologize for not presenting temperature values earlier. To answer the reviewer's question, we start with the first part of the question regarding the physical model and numerical results of the temperature distribution in VO₂ during current induced IMT.

Simulations of temperature were performed for a VO₂ device of 4 μm (width) × 2 μm (length) under different applied currents of 50, 100 and 140 μA (Fig. R32), assuming that the device remains in the insulating phase. In each case, the temperature is highest at the center of the active region and slightly lower near gold electrodes, as evident in the line cuts through electrodes and the active region. This lowered temperature at the edges to electrodes can be explained by the facilitated heat dissipation through gold electrodes. Notably, the highest temperature in the active region rises above 308 K under an applied current of 140 μA, thus enabling the current induced IMT, which agrees with experiment. In reality, the current induced

IMT coincides with release of a significant amount of heat and further rise of temperature [7] which is not accounted in our simulations.

Figure R32: Simulated steady-state temperature distribution under different magnitudes of total applied current. **a**, 50 μA . **b**, 100 μA . **c**, 140 μA . **d**, Line profiles corresponding to the position of the dashed white line in **a**.

COMSOL Multiphysics 6.2 on a workstation was used for finite-element simulations of temperature distributions of a VO_2 device. The model consisted of a TiO_2 substrate, a 10 nm-thick VO_2 film and two Au contact electrodes (40 nm thick each) on top of the film. The film region within the spacing of the electrodes (active region) was 4 μm (width) \times 2 μm (length), same as in experiments.

The physical equations of the model are shown below:

$$\mathbf{J} = -\sigma \nabla V,$$

$$\nabla \cdot \mathbf{J} = 0,$$

$$Q_{\text{Joule}} = -\mathbf{J} \cdot (\nabla V),$$

$$-\nabla \cdot (\kappa \nabla T) = Q_{Joule},$$

where J , V , Q_{Joule} , and T are current density, voltage, Joule heat density and temperature, respectively. σ and κ are electrical and thermal conductivities of the material, respectively.

Each simulation was performed in two steps. First, current was applied between the electrodes and the current density distribution in the film was computed. In the second step, Joule heat distribution was computed based on the solution of the first step, and the steady-state heat transfer in the whole model was simulated to give the temperature distribution. The temperature at the bottom of the TiO₂ substrate was set to 285 K. All other surfaces in the model were set to be electrically and thermally insulating. Tetrahedral domain meshes and triangular surface meshes were used for the finite-element method. The experimental electrical conductivity of VO₂ (insulating state) was used for the film. The thermal conductivities of the materials were taken from literature (accessed from the COMSOL database).

Materials	Electrical conductivity at 285 K	Thermal conductivity at 285 K (W·K ⁻¹ ·m ⁻¹)
Au	Ideal conductor	314 [17]
TiO ₂	Ideal insulator	8.69 [18]
VO ₂	5.0 S·cm ⁻¹	4.4 [19]

Table 1: Summary of materials properties used in the simulations.

We have updated Fig. 3c and d of the manuscript to incorporate the temperature values for the temperature distribution at an applied current 140 μ A. The details of the simulation have been added to the revised Supplementary section 9.

To address the second part of the reviewer's question: we do not have s-SNOM measurements tracking the evolution of the high-temperature region (PeMP) under progressively increasing current after PeMP formation. PeMP nucleation is abrupt, coincident with the current-induced IMT. At lower currents the device enters an oscillatory regime, while applying higher currents poses a risk of device damage. Therefore, we intentionally avoided exceeding the IMT current. Instead, we examined the temperature-driven evolution of the PeMP and found it to be most metallic at the center, expanding radially outward, consistent with our thermal simulations (see Supplementary Section 7 for s-SNOM results without injected current). In addition, s-SNOM imaging on separate devices shows that when the injected current exceeds the IMT threshold

significantly, the resulting PeMP is larger (shown in the Supplementary section S4 for device 4 and 5).

In summary, we have presented our steady state COMSOL simulations on temperature distribution with quantitative values, updated Fig. 3c,d of the manuscript to show the temperature values explicitly for an applied current 140 μA , and discussed our results on the evolution of the high temperature region (PeMP) with temperature and currents exceeding the threshold current for current-induced IMT.

3. VO₂ is sensitive to any thermal fluctuation at the critical temperature. The authors stated that they have keep the laser power as low as possible to eliminate any perturbation. In SI, the power is given by 0.8 mW. Could the authors give more details on experiment parameters such as the spot size? It is suggested that authors make rigorous analysis to show that the heat effect of this light power can be ignored?

Response: We thank the reviewer for this important point. VO₂ is indeed highly sensitive near the insulator-metal transition. Below we quantify the temperature rise for our mid-IR illumination at $\lambda = 10 \mu\text{m}$ with incident power $P_{\text{inc}} = 0.8 \text{ mW}$, both for the near-field (s-SNOM) hotspot ($\sim 40 \text{ nm}$, set by the tip apex) and determines the resolution of the s-SNOM system and for a typical diffraction-limited far-field laser spot.

First, we calculated the absorptance of the VO₂ film which determines how much of the incident power is absorbed by the film:

In our setup, the laser light of wavelength $\lambda = 10 \mu\text{m}$ is p-polarized and incident at an angle of $\theta_i = 60^\circ$. For a VO₂ film with complex refractive index $n_t = n + ik$ and thickness $d = 10 \text{ nm}$ on top of TiO₂ substrate (considered semi-infinite), ignoring interference effects and considering multiple reflections from air-VO₂ and VO₂-TiO₂ surfaces denoted by Fresnel reflectance R_1 and R_2 [20], absorptance can be written as [21] :

$$A_p = 1 - R_p - T_p = \frac{(1 - R_1)(1 - \eta)(1 + R_2\eta)}{1 - R_1R_2\eta^2}$$

Where $\eta = e^{-\frac{\alpha d}{\cos\theta_1}}$ is the one-pass attenuation inside the film with absorption coefficient $\alpha = 4\pi k/\lambda$, transmission angle $\theta_1 = \arcsin\left(\frac{n_i}{n_t} \sin \theta_i\right)$. Using the insulating VO₂ values $n = 2.8$,

$k = 0.15$ [22] and TiO_2 values $n = 2.0$, $k = 0.15$ [23], we obtain $A_p \approx 1.4 \times 10^{-3} \approx 0.14\%$.

Rise in temperature by far-field laser spot:

In the far-field, the absorbed power is:

$$P_{\text{abs}} = P_{\text{inc}} A_p = 0.8 \times 10^{-3} \times 1.4 \times 10^{-3} \approx 1.12 \times 10^{-6} \text{ W}$$

To estimate the rise in temperature, we need to calculate the spot size of the diffraction limited beam. For $\lambda = 10 \mu\text{m}$ and a parabolic mirror with $\text{NA} \approx 0.46$, the diffraction-limited $1/e^2$ diameter is: $D_{\text{FF}} \approx \frac{2\lambda}{\pi \text{NA}} \approx 13.8 \mu\text{m}$ so we take a conservative radius $a = \sqrt{2}\sigma = \frac{D_{\text{FF}}}{2} \approx 7 \mu\text{m}$. According to the Gaussian heat flux model on a semi-infinite substrate, the absorbed heat flux is [24-26]:

$$q(r) = \frac{P_{\text{abs}}}{2\pi\sigma^2} e^{-\frac{r^2}{2\sigma^2}} \text{ Wm}^{-2}$$

Given the thermal conductivities $\kappa_{\text{VO}_2} = 4.4 \text{ Wm}^{-1}\text{K}^{-1}$ and $\kappa_{\text{TiO}_2} = 8.69 \text{ Wm}^{-1}\text{K}^{-1}$, the steady state peak rise at the substrate surface is [26]:

$$\Delta T_{\text{sub}}(0) = \frac{q(0)a}{2\kappa_{\text{TiO}_2}} = \frac{P_{\text{abs}}}{2\pi\kappa_{\text{TiO}_2}a} = \frac{1.12 \times 10^{-6}}{2\pi \times 8.69 \times 7 \times 10^{-6}} \approx 2.9 \text{ mK}$$

Film contribution at the center (added in series) and ignoring the film-substrate interface ($\Delta T_{\text{interface}}$) effect:

$$\Delta T_{\text{film}}(0) = \frac{q(0)d}{\kappa_{\text{VO}_2}} = \frac{P_{\text{abs}}d}{\pi\kappa_{\text{VO}_2}a^2} = \frac{1.12 \times 10^{-6} \times 10 \times 10^{-9}}{\pi \times 4.4 \times 7 \times 10^{-6} \times 7 \times 10^{-6}} \approx 0.017 \text{ mK}$$

Hence, the total temperature rise is 2.92 mK.

Rise in temperature by near-field nanospot:

To calculate the absorbed power in the nano-spot, we calculated the fraction of the incident power falling in this nanospot without considering any near-field enhancement effects. For a tiny disk $r \ll a$, area fraction from FF spot (diameter $13.8 \mu\text{m}$) to tip (diameter 40 nm , $a = 20 \text{ nm}$):

$$f = \left(\frac{40 \text{ nm}}{13.8 \mu\text{m}}\right)^2 \approx 8.4 \times 10^{-6}$$

The absorbed power within the 40 nm disk is $f \times P_{\text{abs}} \sim 0.009$ nW. The Gaussian heat flux model yields a baseline local rise:

$$\Delta T_{\text{sub}}(0) = \frac{P_{\text{abs}}}{2\pi \kappa_{\text{TiO}_2} a} = \frac{0.009 \times 10^{-9}}{2\pi \times 8.69 \times 20 \times 10^{-9}} \approx 0.008 \text{ mK}$$

$$\Delta T_{\text{film}}(0) = \frac{P_{\text{abs}} d}{\pi \kappa_{\text{VO}_2} a^2} = \frac{0.009 \times 10^{-9} \times 10 \times 10^{-9}}{\pi \times 4.4 \times 20 \times 10^{-9} \times 20 \times 10^{-9}} \approx 0.016 \text{ mK}$$

$$\Delta T = \Delta T_{\text{sub}} + \Delta T_{\text{film}} \approx 0.024 \text{ mK}$$

However, a field enhancement at the s-SNOM tip apex cannot be ignored. Therefore, we scale the local intensity by a field-enhancement factor F (one of the experimentally reported value for non-resonant tips [27] was 23) [28]. In this intensity-scaling picture,

$$P_{\text{abs}} \approx F P_{\text{abs},0}, \Delta T \approx F \Delta T_0.$$

For $F = 23$, $\Delta T \approx 0.6$ mK. This value provides an intuitive, field-enhancement based estimate of local heating under the tip.

As an unrealistic upper limit, if we assume that the total power absorbed by the material in the FF spot (1.12×10^{-6} W) can be coupled to the tiny volume fraction ($a = 20$ nm) under the tip due to an extreme field enhancement, the rise in substrate surface temperature would be:

$$\Delta T_{\text{sub}}(0) = \frac{P_{\text{abs}}}{2\pi \kappa_{\text{TiO}_2} a} = \frac{1.12 \times 10^{-6}}{2\pi \times 8.69 \times 20 \times 10^{-9}} \approx 1.03 \text{ K}$$

And the film temperature:

$$\Delta T_{\text{film}}(0) = \frac{P_{\text{abs}} d}{\pi \kappa_{\text{VO}_2} a^2} = \frac{1.12 \times 10^{-6} \times 10 \times 10^{-9}}{\pi \times 4.4 \times 20 \times 10^{-9} \times 20 \times 10^{-9}} \approx 2.03 \text{ K}$$

$$\Delta T = \Delta T_{\text{sub}} + \Delta T_{\text{film}} \approx 3.1 \text{ K}$$

An overestimated laser heating induced temperature rise of 3.1 K is still not enough to cause an IMT at 285 K sample temperature when the warming transition is ~ 308 K. Even at 295 K, the closest to the transition, we still required an electrical bias (≈ 48 μA) to form a conducting element. By contrast, previous reports on laser-heating experiments (at 532 nm, ~ 500 nm spot, ~ 100 nm VO_2 films) typically used >10 mW optical power and reported filament loss when the power dropped below ~ 7.8 mW [15]. At 532 nm, insulating VO_2 has a penetration depth of ~ 125 nm, so a 100 nm film absorbs a substantial fraction of the beam, whereas in our case the

10 μm mid-IR penetration depth is $\sim 5 \mu\text{m}$, and our VO_2 is only 10 nm thick, so only a tiny fraction of the incident mid-IR power is absorbed, yielding negligible heating in the VO_2 film.

In summary, we have calculated the laser induced heating caused by the far-field and near-field spot and found that the negligible rise in the temperature is not enough to cause an IMT in VO_2 and can be effectively ignored. This calculation has been added to the revised Supplementary materials as Section S13.

1. Zhang, J.W., McLeod, A.S., Han, Q., Chen, X.Z., Bechtel, H.A., Yao, Z.Z., Corder, S.N.G., et al., *Nano-Resolved Current-Induced Insulator-Metal Transition in the Mott Insulator Ca_2RuO_4* . *Phys. Rev. X*, 2019. **9**(1) <https://doi.org/10.1103/PhysRevX.9.011032>
2. Jones, A.C., Berweger, S., Wei, J., Cobden, D. and Raschke, M.B., *Nano-optical Investigations of the Metal-Insulator Phase Behavior of Individual VO_2 Microcrystals*. *Nano Letters*, 2010. **10**(5): p. 1574-1581 <https://doi.org/10.1021/nl903765h>
3. Jeong, J., Aetukuri, N.B., Passarello, D., Conradson, S.D., Samant, M.G. and Parkin, S.S.P., *Giant reversible, facet-dependent, structural changes in a correlated-electron insulator induced by ionic liquid gating*. *Proceedings of the National Academy of Sciences of the United States of America*, 2015. **112**(4): p. 1013-1018 <https://doi.org/10.1073/pnas.1419051112>
4. Pang, G.K.H., Baba-Kishi, K.Z. and Patel, A., *Topographic and phase-contrast imaging in atomic force microscopy*. *Ultramicroscopy*, 2000. **81**(2): p. 35-40 [https://doi.org/10.1016/S0304-3991\(99\)00164-3](https://doi.org/10.1016/S0304-3991(99)00164-3)
5. Kim, D.H. and Kwok, H.S., *Pulsed-Laser Deposition of VO_2 Thin-Films*. *Applied Physics Letters*, 1994. **65**(25): p. 3188-3190 <https://doi.org/10.1063/1.112476>
6. Muraoka, Y. and Hiroi, Z., *Metal-insulator transition of VO_2 thin films grown on TiO_2 (001) and (110) substrates*. *Applied Physics Letters*, 2002. **80**(4): p. 583-585 <https://doi.org/10.1063/1.1446215>
7. Li, G., Wang, Z., Chen, Y., Jeon, J.C. and Parkin, S.S.P., *Computational elements based on coupled VO_2 oscillators via tunable thermal triggering*. *Nature Communications*, 2024. **15**(1) <https://doi.org/10.1038/s41467-024-49925-3>
8. Bohaichuk, S.M., Rojo, M.M., Pitner, G., McClellan, C.J., Lian, F.F., Li, J.S., Jeong, J., et al., *Localized Triggering of the Insulator-Metal Transition in VO_2 using a Single Carbon Nanotube*. *Acs Nano*, 2019. **13**(10): p. 11070-11077 <https://doi.org/10.1021/acsnano.9b03397>
9. Bohaichuk, S.M., Kumar, S., Pitner, G., McClellan, C.J., Jeong, J., Samant, M.G., Wong, H.S.P., et al., *Fast Spiking of a Mott VO_2 -Carbon Nanotube Composite Device*. *Nano Letters*, 2019. **19**(10): p. 6751-6755 <https://doi.org/10.1021/acs.nanolett.9b01554>
10. Tsuji, Y., Kanki, T., Murakami, Y. and Tanaka, H., *Single-step metal-insulator transition in thin film-based vanadium dioxide nanowires with a 20 nm electrode gap*. *Applied Physics Express*, 2019. **12**(2) <https://doi.org/10.7567/1882-0786/aafa9e>
11. Yi, W., Tsang, K.K., Lam, S.K., Bai, X., Crowell, J.A. and Flores, E.A., *Biological plausibility and stochasticity in scalable VO_2 active memristor neurons*. *Nature Communications*, 2018. **9** <https://doi.org/10.1038/s41467-018-07052-w>

12. Qazilbash, M.M., Brehm, M., Chae, B.G., Ho, P.C., Andreev, G.O., Kim, B.J., Yun, S.J., et al., *Mott transition in VO₂ revealed by infrared spectroscopy and nano-imaging*. Science, 2007. **318**(5857): p. 1750-1753 <https://doi.org/10.1126/science.1150124>
13. Lahneman, D.J., Slusar, T., Beringer, D.B., Jiang, H.Y., Kim, C.Y., Kim, H.T. and Qazilbash, M.M., *Insulator-to-metal transition in ultrathin rutile VO₂/TiO₂(001)*. Npj Quantum Materials, 2022. **7**(1) <https://doi.org/10.1038/s41535-022-00479-x>
14. Sternbach, A.J., Ruta, F.L., Shi, Y., Slusar, T., Schalch, J., Duan, G.W., McLeod, A.S., et al., *Nanotextured Dynamics of a Light-Induced Phase Transition in VO₂*. Nano Letters, 2021. **21**(21): p. 9052-9060 <https://doi.org/10.1021/acs.nanolett.1c02638>
15. Feng, C., Li, B.W., Dong, Y., Chen, X.D., Zheng, Y., Wang, Z.H., Lin, H.B., et al., *Quantum imaging of the reconfigurable VO₂ synaptic electronics for neuromorphic computing*. Science Advances, 2023. **9**(40) <https://doi.org/10.1126/sciadv.adg9376>
16. Del Valle, J., Vargas, N.M., Rocco, R., Salev, P., Kalcheim, Y., Lapa, P.N., Adda, C., et al., *Spatiotemporal characterization of the field-induced insulator-to-metal transition*. Science, 2021. **373**(6557): p. 907-911 <https://doi.org/10.1126/science.abd9088>
17. Young, H.D., *University Physics*. 7 ed. 1992: Addison Wesley.
18. Kingery, W.D., Francl, J., Coble, R. L. & Vasilos, T. , *Thermal conductivity: X, data for several pure oxide materials corrected to zero porosity*. Journal of the American Ceramic Society, 1954. **37**: p. 107-110
19. Kizuka, H., Yagi, T., Jia, J.J., Yamashita, Y., Nakamura, S., Taketoshi, N. and Shigesato, Y., *Temperature dependence of thermal conductivity of VO₂ thin films across metal-insulator transition*. Japanese Journal of Applied Physics, 2015. **54**(5) <https://doi.org/10.7567/Jjap.54.053201>
20. Palik, E.D., *Handbook of Optical-Constants*. Journal of the Optical Society of America a-Optics Image Science and Vision, 1984. **1**(12): p. 1297-1297
21. Fox, M., *Optical properties of solids*. 2 ed. 2010, USA: Oxford university press Inc., Newyork.
22. Beaini, R., Baloukas, B., Loquai, S., Klemberg-Sapieha, J.E. and Martinu, L., *Thermochromic VO₂-based smart radiator devices with ultralow refractive index cavities for increased performance*. Solar Energy Materials and Solar Cells, 2020. **205** <https://doi.org/10.1016/j.solmat.2019.110260>
23. Siefke, T., Kroker, S., Pfeiffer, K., Puffky, O., Dietrich, K., Franta, D., Ohlídal, I., et al., *Materials Pushing the Application Limits of Wire Grid Polarizers further into the Deep Ultraviolet Spectral Range*. Advanced Optical Materials, 2016. **4**(11): p. 1780-1786 <https://doi.org/10.1002/adom.201600250>
24. Loze, M.K. and Wright, C.D., *Temperature distributions in laser-heated semi-infinite and finite-thickness media with convective surface losses*. Applied Optics, 1998. **37**(28): p. 6822-6832 <https://doi.org/10.1364/Ao.37.006822>
25. Loze, M.K. and Wright, C.D., *Temperature distributions in semi-infinite and finite-thickness media as a result of absorption of laser light*. Applied Optics, 1997. **36**(2): p. 494-507 <https://doi.org/10.1364/Ao.36.000494>
26. Orekhov, A., Rabinskiy, L. and Fedotenkov, G., *Analytical Model of Heating an Isotropic Half-Space by a Moving Laser Source with a Gaussian Distribution*. Symmetry-Basel, 2022. **14**(4) <https://doi.org/10.3390/sym14040650>
27. Atkin, J.M., Berweger, S., Jones, A.C. and Raschke, M.B., *Nano-optical imaging and spectroscopy of order, phases, and domains in complex solids*. Advances in Physics, 2012. **61**(6): p. 745-842 <https://doi.org/10.1080/00018732.2012.737982>
28. Wagner, M., Fei, Z., McLeod, A.S., Rodin, A.S., Bao, W.Z., Iwinski, E.G., Zhao, Z., et al., *Ultrafast and Nanoscale Plasmonic Phenomena in Exfoliated Graphene Revealed*

by *Infrared Pump-Probe Nanoscopy*. Nano Letters, 2014. **14**(2): p. 894-900
<https://doi.org/10.1021/nl4042577>